# Coating Red Phosphor on Green Luminescent Material for Multi-Mode Luminescence and Advanced Anti-Counterfeit Applications

**Jiale Liu, Bo Chen and Qi Zhu \***

Key Laboratory for Anisotropy and Texture of Materials (Ministry of Education), School of Materials Science and Engineering, Northeastern University, Shenyang 110819, China; 20213916@stu.neu.edu.cn (J.L.); 20214098@stu.neu.edu.cn (B.C.)

\* Correspondence: zhuq@smm.neu.edu.cn; Tel.: +86-24-8367-2700

**Abstract:** Traditional fluorescent anti-counterfeiting materials usually exhibit fixed-wavelength excitation patterns and monochromatic luminescence, which are extremely easy to be counterfeited and have low security. Therefore, there is an urgent need to develop multi-mode fluorescent materials with enhanced security to address this issue. Here, $SrAl_2O_4$:1%Eu,2%Dy@$Y_2O_3$:$Eu^{3+}$ core-shell structured phosphors were prepared via a sol-gel method. Coating $SrAl_2O_4$:Eu,Dy with $Y_2O_3$:$Eu^{3+}$ red phosphor did not significantly change the crystal structure of $SrAl_2O_4$. Under UV excitation at 254 nm, $SrAl_2O_4$:1%Eu,2%Dy@$Y_2O_3$:$Eu^{3+}$ exhibited red emission at 613 nm ($^5D_0 \rightarrow ^7F_2$ transition of $Eu^{3+}$), and a strong green afterglow was observed after removing the UV irradiation. However, blue-green emission at 496 nm was observed under UV excitation at 365 nm, followed by green afterglow upon removal of the light source. Varying the content of the $Y_2O_3$:$Eu^{3+}$ shell yielded different emissions and afterglows. The prepared samples are sensitive to the excitation wavelength and duration and have multimodal luminescence properties, which can be used for anti-counterfeiting patterns. The outcomes in this work indicate that the phosphor is a promising fluorescent material for anti-counterfeiting.

**Keywords:** anti-counterfeiting; $Y_2O_3$:$Eu^{3+}$; core-shell; green afterglow

## 1. Introduction

Counterfeit products have penetrated many industries, including currency and luxury goods [1], posing a significant threat to the global market and greatly harming the rights and interests of citizens, as well as the reputation of the government. To address the problem of counterfeit products, many measures have been taken in response, including radio frequency identification (RFID) [2], magnetic responses [3], laser holograms [4] and so on. However, the high manufacturing cost and ease of imitation have limited the development of these technologies [5]. Fluorescent anti-counterfeiting technology is widely used due to its advantages of low cost, strong identification capability and low toxicity. However, most traditional fluorescent anti-counterfeiting materials usually exhibit fixed-wavelength-excitation modes and monochromatic luminescence, which makes them vulnerable to counterfeiting. Therefore, there is an urgent need to develop multi-mode fluorescent materials with enhanced security to address this issue.

There is a wide variety of optical materials available on the market, including organic dye molecules [6], semiconductor quantum dots [7], carbon dots [8,9], persistent nano-materials [10] and metal-organic frameworks (MOFs) [11]. These materials are increasingly used for product anti-counterfeiting marking. However, the mass production of metal-organic framework (MOF) anti-counterfeiting through chemical synthesis is still a difficult point because the synthesis cost of MOFs is high and the yield is low, and

semiconductor quantum dots have certain toxicity [12]. Therefore, our group decided to simplify the problem by starting with the three primary colors of light (red, green and blue) and searching for suitable raw materials to be used in the preparation of optical anti-counterfeiting materials. Long afterglow luminescent materials are a type of light-storage luminescent material with a unique characteristic. They can slowly release energy in the form of light after the light source is turned off [13]. This slow release can generally last for hours or even days. Currently, the popular long afterglow materials available in the market mainly emit a blue-green afterglow. However, the human eye has an unequal number of cone cells responsible for perceiving the three primary colors. The greater the number of cone cells that perceive different colors, the more sensitive the perception of such colors. Research studies have shown that the human eye is most sensitive to green, followed by red, and least sensitive to blue [14]. Therefore, our research group decided to use long green afterglow luminescent materials and red phosphors to develop multi-mode fluorescent materials. Green-red multimode fluorescence materials are obviously not the only strategy for optical anti-counterfeiting. For instance, multilevel emission at NIR I, NIR II and NIR III is also possible [15]. The advantages and disadvantages regarding these strategies are not a major part of the discussion in this paper and thus will not be repeated.

SrAl$_2$O$_4$:Eu,Dy is a type of long green afterglow luminescent material with various luminescent properties, such as a suitable luminescent color, no radiation, high initial luminescent intensity and long duration (up to 16 h) [16]. Y$_2$O$_3$ doped with Eu$^{3+}$ ions (Y$_2$O$_3$:Eu$^{3+}$) is considered one of the best red phosphors because of its simple chemical composition, high luminescence efficiency, high color purity and long-term stability [17]. Due to the typical luminescent characteristics of Y$_2$O$_3$:Eu$^{3+}$ and SrAl$_2$O$_4$:Eu,Dy, multi-mode luminescence can be achieved by using these two phosphors. Herein, a core-shell structured material of SrAl$_2$O$_4$:Eu,Dy@Y$_2$O$_3$:Eu$^{3+}$ has been prepared. Instead of simply mixing the two phosphors, a sol-gel coating strategy [18] has been used. This allows for a single particle to output multiple emissions and afterglows, which is beneficial for high-level optical anti-counterfeiting.

## 2. Experimental Section

### 2.1. Synthesis of SrAl$_2$O$_4$:1%Eu,2%Dy

The SrAl$_2$O$_4$:1%Eu,2%Dy was synthesized using a conventional high-temperature solid-phase method. Analytical grade reagents, including SrCO$_3$, Al$_2$O$_3$, H$_3$BO$_3$, Y$_2$O$_3$, Eu$_2$O$_3$ and Dy$_2$O$_3$ (the purity of all the materials was 99.99%), were used as starting materials. The first three raw materials were purchased from Sinopharm Chemical Reagent Co., Ltd. (Shanghai, China), and the remaining three were purchased from Guangdong Huizhou Rare-Chem. Hi-Tech. Co., Ltd. (Huizhou, China). And then these reagents were uniformly mixed according to the stoichiometric ratio. Then, the powder was scorched in a hydrogen tube furnace, and the powder was placed in the furnace under the atmosphere of hydrogen and nitrogen at 1350 °C for about 2 h. The scorching was stopped when the temperature in the furnace was reduced to 650 °C. The samples were cooled naturally to room temperature, and the bulk solids were crushed and ground into a fine powder for further analysis.

### 2.2. Preparation of Nitrate Solutions

Eu$_2$O$_3$ and Y$_2$O$_3$ powders were added to ultrapure water, and 25 mL of HNO$_3$ was added drop by drop with a burette. The mixture was stirred in a water bath until the powders were completely dissolved. An excess of acid was added to accelerate the reaction rate. When the solution became colorless and transparent, ultrapure water was added to a total of 200 mL. Turn on the heating button, and the liquid was heated to a volume of 150 mL at a temperature of 98 °C in order to remove HNO$_3$.

### 2.3. Formation of Core-Shell Structures via a Sol-Gel Method

The fluorophores were prepared using an organic polymer viscosifier. Firstly, an appropriate amount of citric acid (CA) was added into a beaker and stirred until the added citric acid was completely dissolved in deionized water, and then, $Y(NO_3)_3$ and $Eu(NO_3)_3$, required for each component, were added sequentially and stirred for half an hour. Then, the configured polyvinyl alcohol solutions (hereinafter abbreviated as PVA, used as an organic polymer viscosifier) of varying mass concentrations (1%, 3%, 5%, 7%) were poured inside the prepared solution and stirred on a magnetic mixer to form a gel; then, $SrAl_2O_4$:1%Eu,2%Dy powder was added and continuously stirred for about 30 min. Then, the gel was centrifuged in a low-speed centrifuge and subsequently dried at 60 °C for 24 h. Next, the mixture was ground in a mortar and fired in a muffle furnace. The powder was placed in the furnace under an air atmosphere and fired at 800 °C for 2 h. The firing process was halted when the temperature in the furnace was reduced to 650 °C. The sample was cooled naturally to room temperature, and then, the bulk solids were crushed and ground into powder for further analysis. The formed samples will be replaced by xY (x = 1, 3, 5, 7) in subsequent plots of the analytical data.

### 2.4. Preparation of Anti-Counterfeiting Logo

The production of the anti-counterfeiting mark was produced using computer numerical control finishing by machining a 2 mm-thick digital pattern template on an 8 mm-thick rectangular aluminum alloy surface. Then, the powder, possessing a core-shell structure prepared by an organic polymer viscosifier, was uniformly spread in the designed grooves on the template. Finally, a security pattern in the shape of "1923" was created by placing four different concentrations of PVA-coated $SrAl_2O_4$:1%Eu,2%Dy powders in different parts of the digital mold, and the security mark was obtained.

### 2.5. Characterization Techniques

Phase identification and the crystal structure were measured via X-ray diffraction (XRD, Model SmartLab, Rigaku, Tokyo, Japan). Diffraction data were examined at 40 kV/40 mA with nickel-filtered Cu Ka (k = 0.15406 nm) radiation. The morphology of the samples was measured using a JEM-2100FX transmission electron microscope (TEM, JEOL, Tokyo, Japan). The photoluminescence and fluorescence decay curves was measured using a fluorescence spectrophotometer (model FP-8600, Jasco, Tokyo, Japan) with a 150 W Xe-lamp as the excitation source. The slit widths on both the excitation and emission sides were 10 nm. The diffuse reflectance spectra of the samples were measured with a UV-Vis-NIR spectrophotometer (UV-3600 Plus, Shimadzu, Kyoto, Japan) at room temperature.

## 3. Results and Discussion

### 3.1. Phase Identification and Crystal structure Characterization

Typical X-ray diffraction patterns of the synthesized $SrAl_2O_4$:Eu,Dy@$Y_2O_3$:$Eu^{3+}$ are shown in Figure 1, and the XRD patterns of the parent materials $SrAl_2O_4$ and $Y_2O_3$ are provided for comparison. Figure 1 shows the XRD patterns of the samples formed by coating $Y_2O_3$:$Eu^{3+}$ with different concentrations of PVA (1%, 3%, 5% and 7%) on the surface of $SrAl_2O_4$:Eu,Dy. The diffraction peaks at $2\theta$ = 19.9°, 20.1°, 22.7°, 28.4°, 29.3°, 29.9°, 34.8°, 35.1°, 42.9°, 46.5°, 47.1°, 60.2° and 62.6° are attributed to $SrAl_2O_4$ (PDF#74-0794) and correspond to the (0 1 1), (0 2 0), (1 2 0), (−2 1 1), (2 2 0), (2 1 1), (0 0 2), (0 3 1), (4 0 0), (2 4 0), (2 2 2), (2 4 2) and (5 2 1) planes of $SrAl_2O_4$, respectively. The diffraction peaks at $2\theta$ = 29.2°, 33.8°, 48.5° and 57.6° are attributed to $Y_2O_3$ (PDF#86-1326) and correspond to the (2 2 2), (4 0 0), (4 4 0) and (6 2 2) planes of $Y_2O_3$, respectively. This result confirms that the core-shell particles are composed of both $SrAl_2O_4$ and $Y_2O_3$ phases. It is noteworthy that small amounts of the doped rare-earth reactive ions $Dy^{3+}$ and $Eu^{3+}$ have little effect on the basic crystal structures of $SrAl_2O_4$ and $Y_2O_3$. The XRD patterns show that $SrAl_2O_4$:Eu,Dy and

$Y_2O_3:Eu^{3+}$ have good crystallinity, and the diffraction peaks of these two phases do not change significantly with an increase in the PVA concentration, which indicates that PVA only plays the role in viscosity enhancement, and it has no effect on the basic crystal structures of $SrAl_2O_4$ and $Y_2O_3$. The manifestations of a narrower peak width, sharper peak shape and higher signal-to-noise ratio indicate that the sample preparation process is excellent without the influence of excessive lattice defects and impurities. However, the PVA changes the free energy of different crystal faces, thus changing its growth rate [19], resulting in different XRD diffraction peaks.

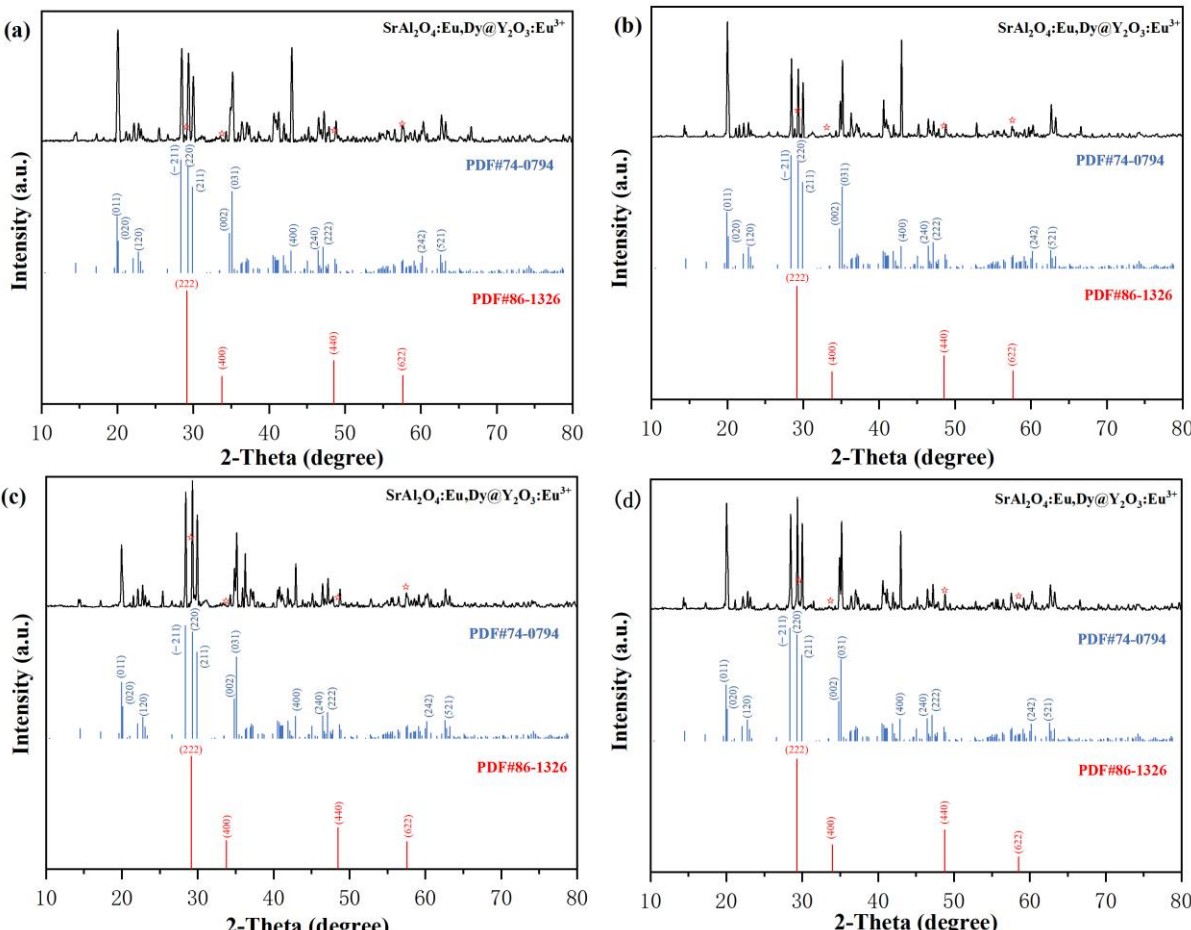

**Figure 1.** X-ray diffraction patterns of $SrAl_2O_4:Eu,Dy@Y_2O_3:Eu^{3+}$ synthesized with different concentrations of PVA: (**a**) 1%, (**b**) 3%, (**c**) 5%, (**d**) 7%. ☆ representative positions of the $Y_2O_3$ peaks.

Figure 2a,b shows the TEM morphology of $SrAl_2O_4:Eu,Dy@Y_2O_3:Eu^{3+}$. Obviously, the samples are irregular micron particles with small nano particles on the surface, indicating that $SrAl_2O_4:1\%Eu,2\%Dy$ has been coated with $Y_2O_3:Eu^{3+}$. We further analyzed the structure and crystallinity of $SrAl_2O_4:Eu,Dy@Y_2O_3:Eu^{3+}$ with the corresponding HR-TEM images (Figure 2c) produced using a high-resolution transmission electron microscope, and the high crystallinity of the samples was confirmed based on the clear punctate streaks in the images. The measured facet spacing is 0.313 nm, corresponding to the (222) crystal plane of $Y_2O_3$, and the measured facet spacing agrees with the calculated results, further indicating $Y_2O_3:Eu^{3+}$ on the surface of $SrAl_2O_4:Eu,Dy$. The selected area electron diffraction (SAED) patterns inset in Figure 2b clearly reveal the (222), (400), (440) and (622) planes of $Y_2O_3$. Figure 2d shows a schematic of the structure of $SrAl_2O_4:Eu,Dy@Y_2O_3:Eu^{3+}$ with $Y_2O_3:Eu^{3+}$ as the shell and $SrAl_2O_4:Eu,Dy$ as the core. The function of this structure is to emit red light during excitation and produce a long green afterglow after excitation. The structure is favorable to improve the luminous intensity and stability because the shell

layer can protect the core layer from the influence of the external environment. The molar ratio of Y: Sr (R) is analyzed via chemical analysis, assuming that the Y and Sr are from $Y_2O_3$:$Eu^{3+}$ and $SrAl_2O_4$:Eu,Dy respectively. Increasing the concentration of PVA from 1% to 7% yields an increased R value from ~0.011 to ~0.083, indicating more $Y_2O_3$:$Eu^{3+}$ on $SrAl_2O_4$:Eu,Dy.

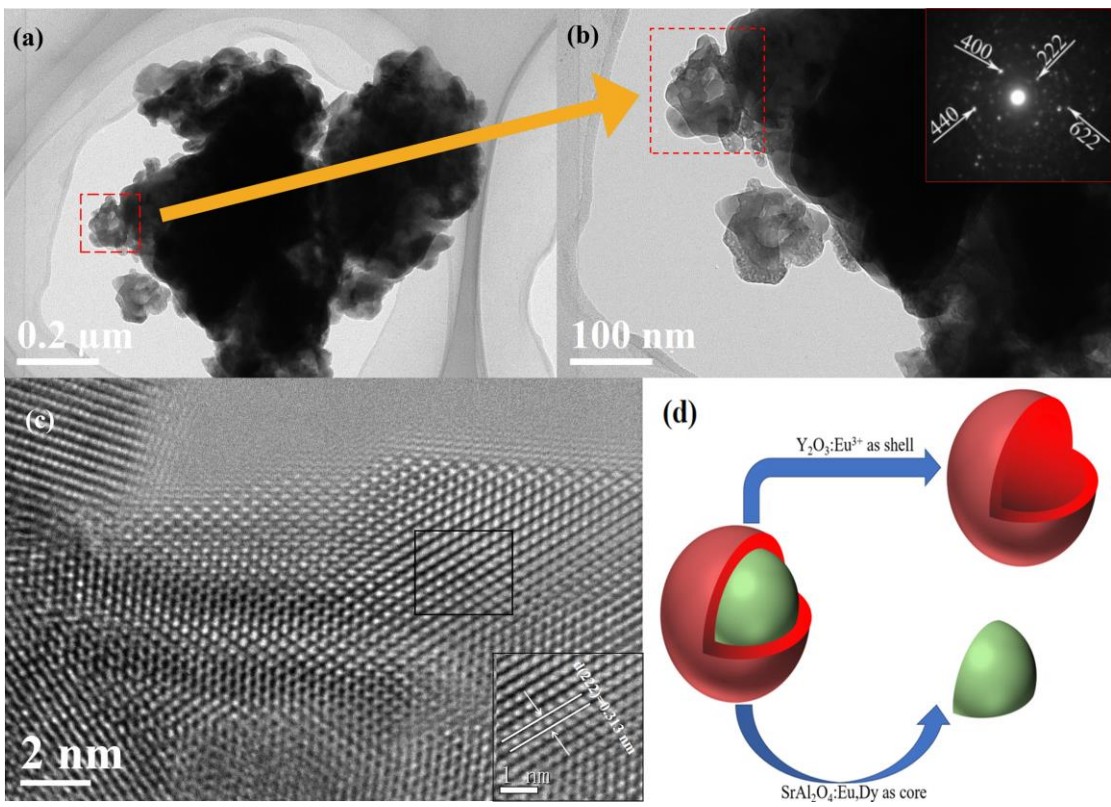

**Figure 2.** (**a**,**b**) TEM, and (**c**) HR-TEM images of $SrAl_2O_4$:Eu,Dy@$Y_2O_3$:$Eu^{3+}$. (**d**) Schematic diagram of the core-shell structure of $SrAl_2O_4$:Eu,Dy@$Y_2O_3$:$Eu^{3+}$.

### *3.2. Photoluminescence of the Phosphors*

Figure 3a–d shows the PLE (Figure 3a,c) and PL spectra (Figure 3b,d) of $SrAl_2O_4$:Eu,Dy@$Y_2O_3$:$Eu^{3+}$ particles at four PVA concentrations (1%, 3%, 5%, 7%), respectively. The broad and strong PLE bands (maximum at 254 nm) can be attributed to charge transfer leaps, and the largest strong excitation band at 254 nm is the charge transfer (CT), i.e., the electronic transition from the 2p orbital of $O^{2-}$ to the 4f orbital of $Eu^{3+}$ (Figure 3a) [20]. That is, under UV excitation at 254 nm, the phosphor exhibits characteristic emission of $Eu^{3+}$ in the cubic $Y_2O_3$ host lattice [17]. The strongest emission peak near 613 nm was due to the forced electronic dipole $^5D_0 \rightarrow {}^7F_2$ transition of $Eu^{3+}$. As marked in Figure 3b, other emission bands were observed corresponding to the $^5D_0 \rightarrow {}^7F_0$ transition at 582 nm, the magnetic dipole $^5D_0 \rightarrow {}^7F_1$ transition at 589, 595 and 601 nm (three Stark splits) and the $^5D_0 \rightarrow {}^7F_3$ transition at 652 nm [17,18,21,22]. The cubic $Y_2O_3$ lattice is known to have two different types of crystallographic positions for $Eu^{3+}$ substitution: one is the 24d site with $C_2$ symmetry and the other is the 8b site with $S_6$ inversion symmetry. The ratio of $C_2$ to $S_6$ is 3:1 (75% and 25% of the total $Y^{3+}$ ions occupied the $C_2$ and $S_6$ sites, respectively), and the $Eu^{3+}$ activators are expected to occupy these two sites in a statistical way upon replacing $Y^{3+}$ [17,22]. According to the Judd–Ofelt parity law [23,24], the magnetic dipole transition is permitted for f-f transitions due to the even parity of the magnetic dipole operator, while the electric dipole transition is allowed only when the $Eu^{3+}$ ions occupy sites in the absence of an inversion center, thus obtaining the mixed states of 4f states (odd parity) and d-f or f-g states (even parity). As the $C_2$ site has no center of inversion and has a statistically

higher occupancy level (75%), the whole emission spectrum is thus dominated by the 613 nm red emission arising from $Eu^{3+}$ ions occupying the $C_2$ sites [17,18,21,22]. Obviously, the emission intensity at 613 nm increases at a higher concentration of PVA, because of more $Y_2O_3{:}Eu^{3+}$ with increased PVA. Meanwhile, a broad emission band at 496 nm is also observed under the excitation of 254 nm, indicating the possible energy transfer from $Eu^{3+}$ to $Eu^{2+}$ that drives the charge transfer from the fundamental energy level of the $4f^7$ structure of $Eu^{2+}$ to the VB [1,25]. From Figure 3c,d, we can see that the optimum emission wavelength of the sample under 365 nm excitation is 496 nm. The intensity of the optimum emission wavelength under 365 nm excitation is not directly proportional to the concentration of PVA, and the strongest emission peak of $Eu^{2+}$ is found at a 5% concentration of PVA. The green emission decreased when the concentration of PVA was increased from 5% to 7%. Mainly due to the energy transfer from $Eu^{3+}$ to $Eu^{2+}$, more $Y_2O_3{:}Eu^{3+}$ with increased PVA resulted in higher emission intensity of $Eu^{2+}$ at 496 nm. However, the increased shell of $Y_2O_3{:}Eu^{3+}$ also contributes to the decreased green emission because of the occlusion of light. Therefore, the optimal green emission intensity of the sample is observed with a 5% PVA concentration. Figure 3e shows the CIE coordinates of the $SrAl_2O_4{:}Eu,Dy@Y_2O_3{:}Eu^{3+}$ samples at different excitation wavelengths, where the point inside circle 1 is the 365 nm excitation point and the point inside circle 2 is the 254 nm excitation point. Under the excitation at 365 nm, the color of the emitting light changes from blue-green to green with an increase in the PVA concentration. While under 254 nm excitation, the emission light changes from yellow-green to orange-red with variation in the PVA concentration.

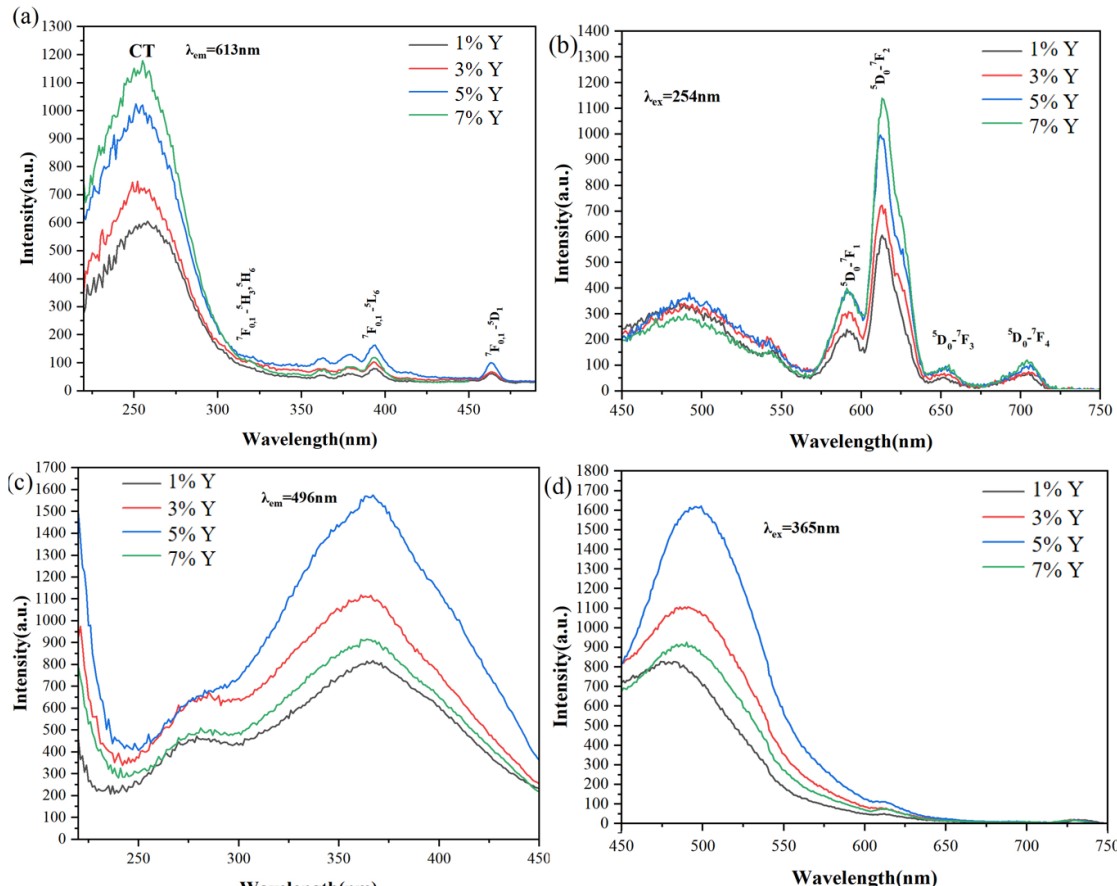

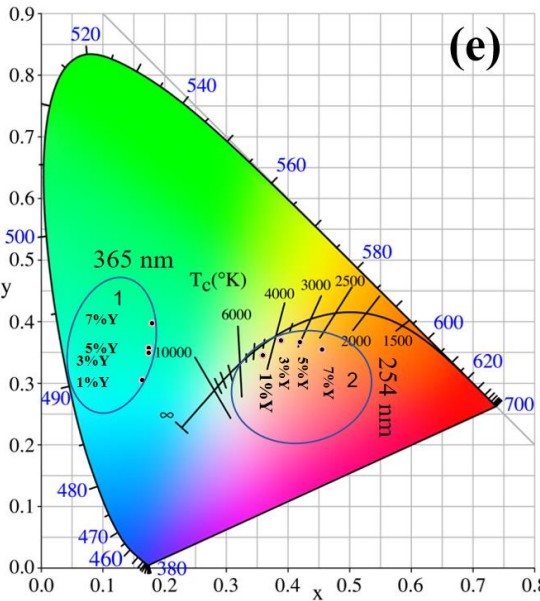

**Figure 3.** (**a**–**d**) PLE/PL spectra and (**e**) CIE coordinates of SrAl$_2$O$_4$:Eu,Dy@Y$_2$O$_3$:Eu$^{3+}$ synthesized with different concentrations of PVA.

### 3.3. Persistent Luminescence and Energy Transfer of the Phosphors

Figure 4 shows the afterglow decay curves of SrAl$_2$O$_4$:Eu,Dy@Y$_2$O$_3$:Eu$^{3+}$ with different contents of Y$_2$O$_3$:Eu$^{3+}$ on the surface, which were monitored at 496 nm after 15 min of 365 nm UV irradiation at room temperature. In the first few tens of seconds, the afterglow intensity is high, and the decay speed is fast, and the afterglow duration of different samples is relatively consistent. According to previous research [26], the long duration afterglow process is dominated by the recombination process of the electrons, which are thermally released from the electron traps. The electrons from the ground state can be pumped to the excitation state of Eu$^{2+}$ and can be captured in the electron traps under the light excitation. Then, the excited electrons can move to the ground state and produce emission. After removing the excitation source, the electrons in the traps jump to the excitation state of Eu$^{2+}$ through CB and back to ground state 4f$^7$ ($^8$S$_{7/2}$) [26], leading to the output of the long afterglow. The co-doping of Dy$^{3+}$ may only serve as the trap centers to modulate the trap distribution via charge compensation. Based on the data presented in Figure 4, it is evident that when the initial afterglow intensity increases within a certain range as the PVA concentration increases and when the PVA concentration reaches 5%, the initial afterglow intensity reaches the highest value. However, at a higher PVA concentration (7%), the initial afterglow intensity decreases rapidly and becomes even lower than that of the samples with a PVA concentration of 1%. This is also mainly due to the energy transfer from Eu$^{3+}$ to Eu$^{2+}$. More Y$_2$O$_3$:Eu$^{3+}$ with increased PVA resulted in higher afterglow intensity of Eu$^{2+}$ at 496 nm. However, the increased shell of Y$_2$O$_3$:Eu$^{3+}$ also contributes to the decreased green afterglow because of the occlusion of light [27,28].

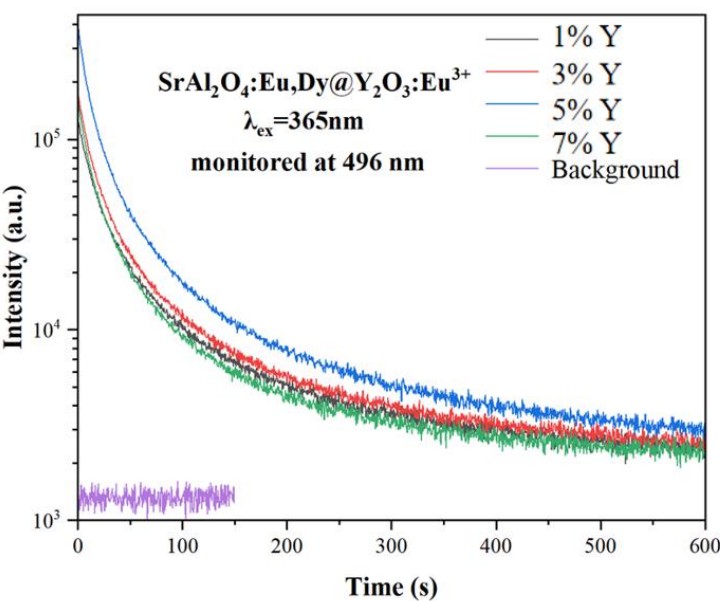

**Figure 4.** Persistent luminescence decay curves for SrAl₂O₄:Eu,Dy@Y₂O₃:Eu³⁺ synthesized with different concentrations of PVA after 254 nm UV light illumination for 15 min. The instrument parameters were kept the same for all the materials.

### 3.4. Multimode Anti-Counterfeiting Applications

Figure 5a,b shows the luminescence and duration of the multimode anti-counterfeiting samples SrAl₂O₄:Eu,Dy@Y₂O₃:Eu³⁺ under 254 nm and 365 nm excitation for five minutes, respectively. The samples prepared under 1%, 3%, 5% and 7% PVA concentration conditions all emitted orange-red light when excited with 254 nm for 5 min. Additionally, a long green afterglow was observed upon de-excitation. But due to variations in the PVA concentration, the four samples exhibit different performances. At a PVA concentration of 7%, the sample generates the strongest red light, but the long green afterglow is the weakest with the shortest duration under this condition. The samples with 5% PVA display the strongest red light intensity, second only to the samples at 7% PVA, and they exhibit the strongest long green afterglow with the longest duration, having the best overall performance in comparison. Under 365 nm excitation conditions, all four samples emit blue-green light under the excitation conditions. After the stoppage of the excitation, they all emit a long green afterglow. However, the difference is that under 5% PVA concentration conditions, the intensity of the blue-green light under the excitation conditions is the highest, and the brightness and duration of the green afterglow after excitation is also the highest. In summary, our samples have successfully achieved the characteristic of multiple emitted light from a single particle, the formation of different colors under the irradiation of different wavelengths of light and the effect of multi-mode fluorescence anti-counterfeiting. In addition, the afterglow emitted by the sample produced under a 5% PVA concentration displays the highest brightness and the longest duration under different excitation conditions.

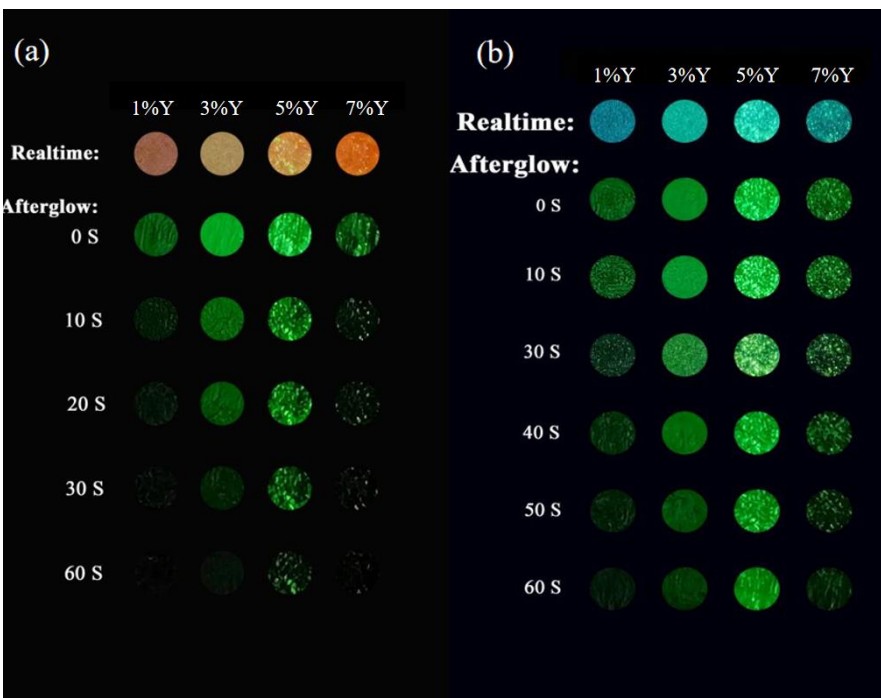

**Figure 5.** Demonstration of the multi-mode anti-counterfeiting SrAl$_2$O$_4$:Eu,Dy@Y$_2$O$_3$:Eu$^{3+}$ samples irradiated at (**a**) 254 nm and (**b**) 365 nm.

Here, we prepared samples with multimodal anti-counterfeiting features and designed a simple pattern to verify their performance. As shown in Figure 6, the pattern consists of four numbers, namely 1, 9, 2 and 3. These four numbers not only represent different numerical information, but also represent different sample preparation conditions. Specifically, each number corresponds to a different polyvinyl alcohol (PVA) concentration, namely 1%, 3%, 5% and 7%. Then, 254 nm and 365 nm ultraviolet lights were used to excite the patterns, and the luminescence characteristics of the samples included real-time and afterglow signals. Figure 6a,b shows the real-time and afterglow photos of the patterns under the excitation of two ultraviolet lights, respectively. From Figure 6a, it can be seen that all samples emit orange-red light at 254 nm excitation, and the pattern is clearly visible. After turning off the ultraviolet lamp, the samples show a long green afterglow, lasting from 20 s to 30 s, among which the sample with a 5% PVA concentration has the strongest afterglow, lasting for about 30 s. From Figure 6b, it can be seen that under the 365 nm excitation, all samples emit blue-green light, and the pattern is also clearly visible. After turning off the ultraviolet lamp, the samples still show a long green afterglow, lasting from 30 s to 60 s, among which the sample with the 5% PVA concentration still has the strongest afterglow, lasting for about 60 s. The above results indicate that our samples have the characteristics of dual-color luminescence, which can realize the switchable color light by turning on and off the excitation light, thereby improving the anti-counterfeiting effect.

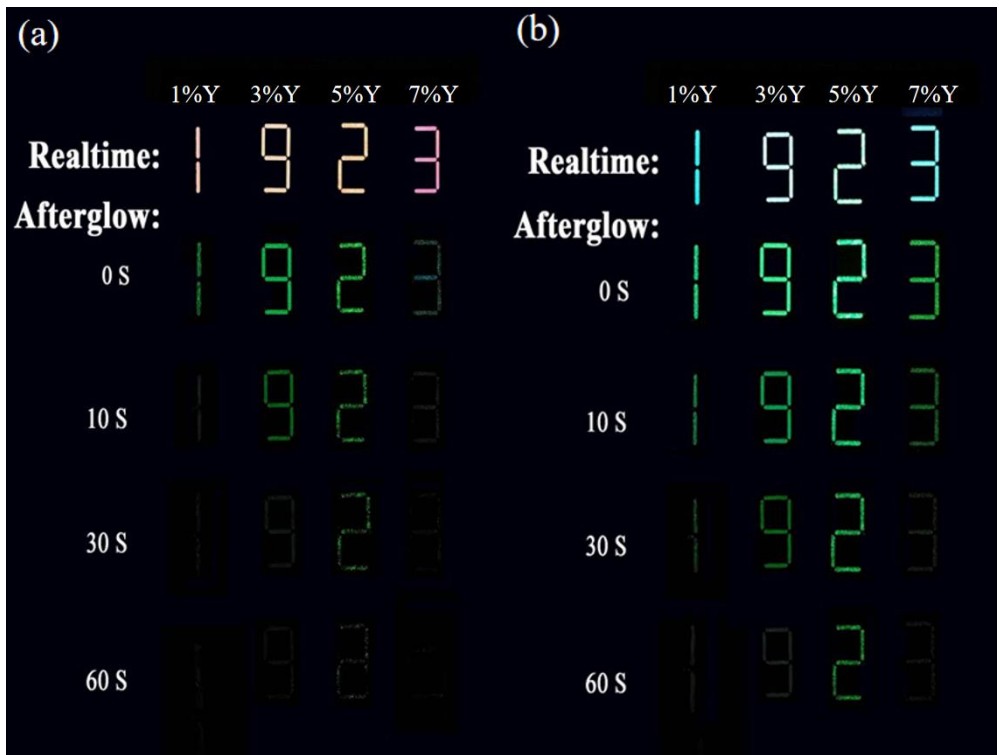

**Figure 6.** Photographs of the logo "1923" under (**a**) 254 nm and (**b**) 365 nm radiation from a box UV lamp and afterglow after the removal of radiation (after 254 nm UV light illumination for 1 min).

## 4. Conclusions

In this work, a novel phosphor with a core-shell structure was successfully synthesized by coating a $Y_2O_3$:$Eu^{3+}$ red phosphor on $SrAl_2O_4$:1%Eu,2%Dy green persistent luminescent materials. XRD, TEM, HR-TEM, PLE/PL spectroscopy and persistent luminescence decay curve analysis were used to characterize samples. Although coating $Y_2O_3$:$Eu^{3+}$ red phosphor onto $SrAl_2O_4$:1%Eu,2%Dy did not significantly change the crystal structure of $SrAl_2O_4$, it did change the luminescence characteristics of the phosphor. $SrAl_2O_4$:1%Eu,2%Dy@$Y_2O_3$:$Eu^{3+}$ exhibited a typical red emission at 613 nm ($^5D_0{\rightarrow}^7F_2$ transition of $Eu^{3+}$) with exposure to a 254 nm UV excitation light. However, removal of the UV irradiation yielded a strong green afterglow. While blue-green emission at 496 nm was observed under UV excitation at 365 nm, green afterglow was detected by removing the light source. Varying the thickness of the $Y_2O_3$:$Eu^{3+}$ shell generated different color signals, including emissions and afterglows. The prepared samples are sensitive to different excitation wavelengths and have multimodal luminescence properties, making the material potentially useful for anti-counterfeiting patterns.

**Author Contributions:** J.L.: data curation, formal analysis, writing—original draft. B.C.: data curation, formal analysis. Q.Z.: resources, supervision, conceptualization, writing—review and editing, conceptualization. All authors have read and agreed to the published version of the manuscript.

**Funding:** Fundamental Research Funds for the Central Universities (Grant N2302004) and National Natural Science Foundation of China (Grant 52371057).

**Institutional Review Board Statement:** Not applicable.

**Informed Consent Statement:** Not applicable.

**Data Availability Statement:** Data available on request from the authors.

**Conflicts of Interest:** The authors declare that they have no known competing financial interests or personal relationships that could have appeared to influence the work reported in this paper.

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
