# Peer review of "Coating Red Phosphor on Green Luminescent Material for Multi-Mode Luminescence and Advanced Anti-Counterfeit Applications"

_coatings, doi:10.3390/coatings14040509_

Round 1
Reviewer 1 Report
Comments and Suggestions for Authors
Document attached.

Author Response
Minor comments:
- Page 1, line 26, Introduction: “Counterfeit products have become penetrated in many industries…”
Answer: The mentioned issue has been revised. Please see page 1, line 26. Thank you.
- Page 1, lines 33-34, Introduction: “Fluorescent anti-counterfeiting technology is widely used due to its advantages of low cost, strong identification capability and non-toxicity.”
I would not say here that one of the advantages of fluorescent anti-counterfeiting compared to other technologies such as radio frequency, magnetic response or laser holograms is non-toxicity as any of these technologies is toxic. And in any case, toxicity is addressed afterwards when comparing persistent luminescence nanoparticles and quantum dots.
Answer: Thank you very much for your constructive comments. Indeed, it is not very accurate to say that it is non-toxicity. The mentioned issue has been revised in page 1, line 33.
- Page 1, lines 41-43, Introduction: “However, mass production of metal-organic frameworks (MOF) anti-counterfeiting through chemical synthesis is still a difficult point, …”
This affirmation is a bit vague; could the authors explain why is a “difficult point”?
Answer: Thank you very much for your constructive comments. Here, the synthesis cost of MOFs is high and the yield is low. The mentioned issue has been revised in page 1, line 43.
- Page 2, lines 46-49, Introduction: “Long afterglow luminescent materials are a type of light storage luminescent material with unique characteristic. They can slowly release energy in the form of light after the light source is turned off. This slow release can generally last for hours or even days”
Some references are needed for the persistent luminescence materials definition.
Answer: Thank you very much for the mentioned issue. The reference is added in page 2, line 49.
- Page 2, lines 53-54, Introduction: “Research studies have shown that the human eye is most sensitive to green, followed by red and least sensitive to blue.”
A reference is needed.
Answer: Thank you very much for the mentioned issue. The reference is added in page 2, line 55.
- Page 2, lines 54-56, Introduction: “Therefore, our research group decided to use long green afterglow luminescent materials and red phosphors to develop multi-mode fluorescent materials.”
I would mention in this paragraph that green-red multimode fluorescence materials are not the only strategy for optical anti-counterfeiting. For instance, multilevel emission at NIR I, NIR II and NIR III is also possible as in:
“ZGSO spinel nanoparticles with dual emission of NIR persistent luminescence for anti-counterfeiting applications” G Cai, T Delgado, C Richard, B Viana - Materials, 2023
DOI: 10.3390/ma16031132
Answer: Thank you very much for your constructive comments. Multilevel emission at NIR I, NIR II and NIR III is a good anti-counterfeiting strategy. We've added it on page2, lines 57-60.
- Page 2, line 86, Experimental: “The heating button was turned on…” There must be a better way to express this.
Answer: Thank you very much for the mentioned issue. The mentioned issue has been revised in page 2, line 90.
- In general sometimes X-ray is written with capital letter and sometimes with lower case.
Answer: Thank you very much for the mentioned issue. The mentioned issue has been revised in page 3, line 116.
- Page 5, lines 186 - 189, Results: “While a broad emission band at 496 nm is also observed under the excitation of 254 nm, indicating the possible energy transfer from Eu3+ to Eu2+that drives the charge transfer from the fundamental energy level of the 4f7 structure of Eu2+ to the VB.”
A reference is needed.
Answer: Thank you very much for the mentioned issue. The references are added in page 6, line 193.
- Page 7, lines 221 - 223, Results: “…when the PVA concentration reaches 5%, the initial afterglow intensity reaches the highest. However, after this at higher PVA concentration (7%) , the initial afterglow intensity decreases rapidly and becomes even lower than that of the samples with a PVA concentration of 1%.”
Answer: The mentioned issue has been revised. Please see page 7, line 227. Thank you.
- There is a lack of geographical diversity in the references and some extra ones are needed as indicating above.
Answer: Thank you very much for the mentioned issue. We have added three different references.

Reviewer 2 Report
Comments and Suggestions for Authors
The manuscript under review entitled “Coating Red Phosphor on Green Luminescent Material for Multi-Mode Luminescence and Advanced Anti-Counterfeit Applications” deals with synthesis and luminescence properties of coating based on SrAl2O4:Eu,Dy with Y2O3:Eu3+ phosphors. The studied materials have clear practical value, and it described well in the manuscript. At the same time, some discussion and conclusions have no experimental verification. Thus, it is recommended Major Revision.
Here are some comments, that can help to improve the manuscript.
1) Authors said, that SrAl2O4: 1%Eu,2%Dy@Y2O3:Eu3+ core-shell structured phosphors were obtained. However, it is not clear from TEM/HRTEM images that core-shell structures were formed. More likely, there are agglomerations of SrAl2O4: 1%Eu,2%Dy and Y2O3:Eu3+ particles. Maybe there are some TEM images where core and shell of the particle can be clearly seen?
2) What ratio between SrAl2O4: 1%Eu,2%Dy and Y2O3:Eu3+ component at the sample? It looks like from XRD patterns and luminescence results that Y2O3:Eu3+ content is not very high.
3) Lines 92-95. It is said, “…PVA, used as an organic polymer viscosifier) of varying mass concentrations (1%, 3%, 5%, 7%) was poured inside the prepared solution”. From this phrase it can be concluded that with increasing of PVA content the quantity of Y2O3:Eu in the samples should be smaller. Later (lines 184-186) authors said “the emission intensity at 613 nm increases at a higher concentration of PVA, because of more Y2O3:Eu3+ with increased PVA.” It is not clear how it can be more Y2O3:Eu3+ with increased PVA content? Specify a content of SrAl2O4: 1%Eu,2%Dy, Y2O3:Eu3+, and PVA in each sample.
4) The peaks of Y2O3 phase are hardly seen from XRD patterns (Fig 1). Please show positions of the Y2O3 peaks on experimental patterns by arrows or asterisks.
5) Figure 2c. Authors said, that “…facet spacing correspond to (222) crystal plane of Y2O3”. In my opinion, this facet spacing can be ascribed rather to (-2-11) plane in monoclinic SrAl2O4 with d=0.314 nm.
6) It is seen from photoluminescence spectra (Fig.3) that Eu3+-related bands are very wide. Such situation is typical for Eu3+ in amorphous phase. Maybe this result is because of large slits were used for measurements? I recommend to measure spectra with smaller slits, e.g. 2 nm, in order to check if these bands became much narrower. Otherwise, it is doubtful that Y2O3 phase has been crystallized.
7) I suggest adding luminescence and its excitation spectra for Y2O3:Eu3+ in Fig. 3 (a,b) and SrAl2O4: 1%Eu,2%Dy in Fig. 3(c,d). Authors can also measure luminescence spectra of PVA and include them in Fig. 3.
8) Band of 5D0-7F0 transition cannot be seen at Fig. 3b, so its denotation has no sense and should be removed.
9) It is better to make Fig. 4 in semilogarithmic scale (i.e., x = t; y = lg(I)) and calculate decay time constants.
Author Response
Reply to Reviewer’ Comments
Here are some comments, that can help to improve the manuscript.
1) Authors said, that SrAl2O4: 1%Eu,2%Dy@Y2O3:Eu3+ core-shell structured phosphors were obtained. However, it is not clear from TEM/HRTEM images that core-shell structures were formed. More likely, there are agglomerations of SrAl2O4: 1%Eu,2%Dy and Y2O3:Eu3+ particles. Maybe there are some TEM images where core and shell of the particle can be clearly seen?
Answer: Thank you very much for your constructive comments. Because the fluorescent particles are irregular, the coating effect is really not very good. But by TEM, it is found that there are fine nanoparticles on the surface of SrAl2O4: 1%Eu,2%Dy, and it is proved to be Y2O3:Eu3+. Therefore, it is not very accurate to say that it is a core-shell structure.
2) What ratio between SrAl2O4:1%Eu,2%Dy and Y2O3:Eu3+ component at the sample? It looks like from XRD patterns and luminescence results that Y2O3:Eu3+ content is not very high.
Answer: Thank you very much for your constructive comments. The exact ratio between SrAl2O4:1%Eu,2%Dy and Y2O3:Eu3+ component at the sample in unclear. Here, the prepared materials are SrAl2O4:1%Eu,2%Dy@Y2O3:Eu3+ core-shell structured phosphors. PVA is a binder used to increase the content of Y2O3:Eu3+ in the shell. As PVA increases, the viscosity of the solution increases, so more Y2O3:Eu3+ is coated on the particle of SrAl2O4:Eu,Dy. The more intense red emission of Y2O3:Eu3+ is indeed found.
3) Lines 92-95. It is said, “…PVA, used as an organic polymer viscosifier) of varying mass concentrations (1%, 3%, 5%, 7%) was poured inside the prepared solution”. From this phrase it can be concluded that with increasing of PVA content the quantity of Y2O3:Eu in the samples should be smaller. Later (lines 184-186) authors said “the emission intensity at 613 nm increases at a higher concentration of PVA, because of more Y2O3:Eu3+ with increased PVA.” It is not clear how it can be more Y2O3:Eu3+ with increased PVA content? Specify a content of SrAl2O4: 1%Eu,2%Dy, Y2O3:Eu3+, and PVA in each sample.
Answer: PVA is a binder used to increase the content of Y2O3:Eu3+ in the shell. After calcination at a high temperature, PVA decomposed and disappeared. As PVA increases, the viscosity of the solution increases, so more Y2O3:Eu3+ is coated on the particle of SrAl2O4:Eu,Dy. More Y2O3:Eu3+ would contribute to more an intense red emission.
4) The peaks of Y2O3 phase are hardly seen from XRD patterns (Fig 1). Please show positions of the Y2O3 peaks on experimental patterns by arrows or asterisks.
Answer: Because of the low content of yttrium oxide, it is not easy to find in XRD. However, we still came out the yttrium oxide phase according to the review opinions.
5) Figure 2c. Authors said, that “…facet spacing correspond to (222) crystal plane of Y2O3”. In my opinion, this facet spacing can be ascribed rather to (-2-11) plane in monoclinic SrAl2O4 with d=0.314 nm.
Answer: It is indeed in the picture the facet spacing correspond to (222) crystal plane of Y2O3, because the SAED data confirms this results. The SAED data is added in the text.
6) It is seen from photoluminescence spectra (Fig.3) that Eu3+-related bands are very wide. Such situation is typical for Eu3+ in amorphous phase. Maybe this result is because of large slits were used for measurements? I recommend to measure spectra with smaller slits, e.g. 2 nm, in order to check if these bands became much narrower. Otherwise, it is doubtful that Y2O3 phase has been crystallized.
Answer: Thank you very much for your constructive comments. Here, the crystallization degree of yttrium oxide is really not high, most of which are nanoparticles, and some amorphous substances may also exist. But SAED results show that crystalline yttrium oxide does exist. Of course, it is true that the emission peak of Eu3+ is broadened due to the low crystallization degree.
7) I suggest adding luminescence and its excitation spectra for Y2O3:Eu3+ in Fig. 3 (a,b) and SrAl2O4: 1%Eu,2%Dy in Fig. 3(c,d). Authors can also measure luminescence spectra of PVA and include them in Fig. 3.
Answer: After calcination at a high temperature, PVA decomposed and disappeared. Here, the prepared materials are SrAl2O4:1%Eu,2%Dy@Y2O3:Eu3+ core-shell structured phosphors. Y2O3:Eu3+ is coated on the particle of SrAl2O4:Eu,Dy. The emissions of Eu3+ in Y2O3 and Eu2+ in SrAl2O4 are shown in Figure. There is no need to put separate emission spectra in this paper, and there is no goal to synthesize them in this paper.
8) Band of 5D0-7F0 transition cannot be seen at Fig. 3b, so its denotation has no sense and should be removed.
Answer: The mentioned issue has been revised. Thank you.
9) It is better to make Fig. 4 in semilogarithmic scale (i.e., x = t; y = lg(I)) and calculate decay time constants
Answer: The mentioned issue has been revised. Thank you.

Reviewer 3 Report
Comments and Suggestions for Authors
There are a large number of inconsistencies in the work:
1.- It is not explained what the effect of adding PVA to the system is, nor the reason why the luminescent properties can change depending on its content. The authors explain in XRD that it does not have any structural effect, but a luminescent effect? Obviously due to the treatment temperature there is no longer PVA in the system, so what do I modify? These compounds are usually added to increase the rate of chain formation in the chemical part of the reaction and thus modify the morphology, for example, but in the work it is not at all evident what the effect of PVA is. Without this part, the entire work becomes invalid. What changed the PVA?
2.- The XRD results are unreliable, since initially the authors mention that PVA has no effect on the structure, which is false, analyzing the differences between the relative intensity ratios between the diffraction maxima. For example, in plane 011, sometimes it is larger than the reference and sometimes it is not. This happens with several signals. This, although initially it could indicate a certain degree of preferential structuring of the system, does not seem to have a direct relationship with the PVA content.
3.- On the other hand, the presence of the Y2O3 phase cannot be directly verified with the way the results are arranged, since the diffraction maximum of cubic Y2O3 coincides with that of SrAl2O4. Therefore, a mathematical analysis (for example Rietveld) would be required to verify the above. The same thing regarding the fact that adding Dy or Eu does not affect the structure, it is well known that they will normally distort the lattice, due to the difference in radii. Minimally, comparison with undoped systems would also be required. In general, the XRD study is poor.
4.- It is very difficult to justify the formation of Eu2+ through the proposed process, solely through the presence of H2, there is extensive literature that this operation is not simple nor does it produce great formation of Eu2+. Even more so because the authors seem to want to imply that PVA would cause this effect, which would not make sense.
5.- The phrase: “ the emission intensity at 613 nm increases at a higher concentration of PVA, because of more Y2O3:Eu3+ with increased PVA”, it doesn't make sense, because PVA would produce such an effect?
6.- Where is the Dy luminescence found? What is its luminescent effect?
7.- The study of the decay time is also not clear, for the same reasons mentioned in the previous points. Furthermore, it is necessary to calculate the average lifetime, minimally, to observe which model it fits and try to explain something more clearly.
Author Response
Reply to Reviewer’ Comments
- It is not explained what the effect of adding PVA to the system is, nor the reason why the luminescent properties can change depending on its content. The authors explain in XRD that it does not have any structural effect, but a luminescent effect? Obviously due to the treatment temperature there is no longer PVA in the system, so what do I modify? These compounds are usually added to increase the rate of chain formation in the chemical part of the reaction and thus modify the morphology, for example, but in the work it is not at all evident what the effect of PVA is. Without this part, the entire work becomes invalid. What changed the PVA?
Answer: Thank you very much for your constructive comments. Here, the prepared materials are SrAl2O4:1%Eu,2%Dy@Y2O3:Eu3+ core-shell structured phosphors. PVA is a binder used to increase the content of Y2O3:Eu3+ in the shell. After calcination at a high temperature, PVA decomposed and disappeared. As PVA increases, the viscosity of the solution increases, so more Y2O3:Eu3+ is coated on the particle of SrAl2O4:Eu,Dy. There is no PVA in the final product. Because the crystal structure of SrAl2O4:Eu,Dy and Y2O3:Eu3+ did not change, PVA has little contribution to their crystal structure. However, more Y2O3:Eu3+ would contribute to more an intense red emission.
2.- The XRD results are unreliable, since initially the authors mention that PVA has no effect on the structure, which is false, analyzing the differences between the relative intensity ratios between the diffraction maxima. For example, in plane 011, sometimes it is larger than the reference and sometimes it is not. This happens with several signals. This, although initially it could indicate a certain degree of preferential structuring of the system, does not seem to have a direct relationship with the PVA content.
Answer: Thank you very much for your constructive comments. Indeed, the unclear expression of “crystal structure” and “structure” in the original text leads to misunderstanding of reviewer. The author has checked and revised the full text. Because the crystal structure of SrAl2O4:Eu,Dy and Y2O3:Eu3+ did not change, PVA has little contribution to their crystal structure. More intense diffraction peak of (110) found in XRD is mainly due to the larger crystal size or more perfect degree of crystallization, rather than the change of crystal structure.
3.- On the other hand, the presence of the Y2O3 phase cannot be directly verified with the way the results are arranged, since the diffraction maximum of cubic Y2O3 coincides with that of SrAl2O4. Therefore, a mathematical analysis (for example Rietveld) would be required to verify the above. The same thing regarding the fact that adding Dy or Eu does not affect the structure, it is well known that they will normally distort the lattice, due to the difference in radii. Minimally, comparison with undoped systems would also be required. In general, the XRD study is poor.
Answer: Thank you very much for your constructive comments. Rietveld refinement is an efficient way to identify the crystal structure. However, the prepared materials are SrAl2O4:1%Eu,2%Dy@Y2O3:Eu3+ core-shell structured phosphors, which is a mixture. In addition, nano and amorphous particles exist in the products, which will have a poor impact on the accuracy of analysis. Fluorescence spectra and afterglow spectra have proved the existence of Eu and Dy. Here, the XRD analysis section has been modified.
4.- It is very difficult to justify the formation of Eu2+ through the proposed process, solely through the presence of H2, there is extensive literature that this operation is not simple nor does it produce great formation of Eu2+. Even more so because the authors seem to want to imply that PVA would cause this effect, which would not make sense.
Answer: A large number of literatures show that hydrogen atmosphere is a reducing atmosphere, which can reduce Eu3+ to Eu2+ [such as J. Am. Ceram. Soc., 2016, 99, 183; Inorg. Chem., 2005, 44, 489]. “Then the powder was scorched in a hydrogen tube furnace, and the powder was placed in the furnace under the atmosphere of hydrogen and nitrogen at 1350°C for about 2 hours.” the hydrogen atmosphere is used here.
5.- The phrase: “the emission intensity at 613 nm increases at a higher concentration of PVA, because of more Y2O3:Eu3+ with increased PVA”, it doesn't make sense, because PVA would produce such an effect?
Answer: PVA is a binder used to increase the content of Y2O3:Eu3+ in the shell. After calcination at a high temperature, PVA decomposed and disappeared. As PVA increases, the viscosity of the solution increases, so more Y2O3:Eu3+ is coated on the particle of SrAl2O4:Eu,Dy. There is no PVA in the final product. More Y2O3:Eu3+ would contribute to more an intense red emission.
6.- Where is the Dy luminescence found? What is its luminescent effect?
Answer: Dy3+ acts as an electron trap here. This part has been supplemented in this paper. Please refer to page 7, lines 222-223 in red.
7.- The study of the decay time is also not clear, for the same reasons mentioned in the previous points. Furthermore, it is necessary to calculate the average lifetime, minimally, to observe which model it fits and try to explain something more clearly.
Answer: SrAl2O4: Eu, Dy is a long afterglow material, and its fluorescence lifetime can not be simply calculated and evaluated. The related mechanism has been supplemented in this paper. Please refer to page 7, lines 215-222 in red.
Round 2
Reviewer 2 Report
Comments and Suggestions for Authors
The authors responded to all comments and improved the manuscript under review.
At the same time, according to these answers, the authors themselves are not sure that the obtained particles can be classified as core-shell. Therefore, I recommend using the term "SrAl2O4:1%Eu,2%Dy coated with Y2O3:Eu3+" instead of core-shell particles.
The abbreviation x%PVA in Figures is quite confusing, as (according to the authors' response) the PVA decomposed and disappeared at high temperatures. It can be mistakenly treated that PVA is a part of particle. So, I suggest using some other abbreviation (e.g. 1Y or 7YO/SrAlO) with its specification in section 2.3.
The manuscript can be accepted for publication after these minor corrections.
Author Response
Reviewer 2
At the same time, according to these answers, the authors themselves are not sure that the obtained particles can be classified as core-shell. Therefore, I recommend using the term "SrAl2O4:1%Eu,2%Dy coated with Y2O3:Eu3+" instead of core-shell particles.
Answer: Thank you very much for the mentioned issue. We have modified it according to your suggestions.
The abbreviation x%PVA in Figures is quite confusing, as (according to the authors' response) the PVA decomposed and disappeared at high temperatures. It can be mistakenly treated that PVA is a part of particle. So, I suggest using some other abbreviation (e.g. 1Y or 7YO/SrAlO) with its specification in section 2.3.
Answer: Thank you very much for the mentioned issue. We have amended the labelling in the Figures as required and added its specification in section 2.3.
Reviewer 3 Report
Comments and Suggestions for Authors
1.- Unfortunately, what was requested in the previous report has not yet been clearly explained. Regarding the effect of the PVA content on the luminescence, what was stated in your response, at least contrasted with your experimental methodology, could not be the case: the PVA could not cause a greater number of luminescent particles to adhere to the surface of the core-shel , since the amount of these in the sol that includes PVA does not change (unless the authors specified otherwise), therefore the luminescent effect cannot be attributed to this factor since, in theory, there would be a difference in the form in which they will be found on the surface, but not the quantity of them. If this were the effect, it would be necessary to present the results of the chemical analysis and observe whether this difference actually exists.
2.- Regarding what was established in the answer that the difference in intensities is due to the level of crystallinity, this cannot be in the present case: the difference in heights can, indeed, be due to a material with higher or lower crystallinity, but in that case, all the intensities of the system are affected in the same way, that is, there is no change in the intensity ratios, for example the ratio I(001)/I(220) must remain approximately constant even if it changes the degree of crystallinity. These ratios are clearly not constant when adding different amounts of PVA, so the answer is not correct.
3.- It is still important, for the previous reason, to know the XRD of the sample without PVA, in order to evaluate its effect, otherwise the entire discussion cannot be verified.
4.- The effect of Dy on the luminescence remains unclear, even assuming that it functioned as an electron-trap (which the same authors propose in the manuscript as "probable"), it would be unlikely that its luminescence would not be observed, especially considering that it is usually found when excited at approximately 350 nm, which is why Its effect should be observed at 365 nm. In order to analyze this effect, it is minimally required to search for the Dy emission at 573 nm and from this result to analyze the luminescent results again.
Author Response
Reviewer 3
1.- Unfortunately, what was requested in the previous report has not yet been clearly explained. Regarding the effect of the PVA content on the luminescence, what was stated in your response, at least contrasted with your experimental methodology, could not be the case: the PVA could not cause a greater number of luminescent particles to adhere to the surface of the core-shell , since the amount of these in the sol that includes PVA does not change (unless the authors specified otherwise), therefore the luminescent effect cannot be attributed to this factor since, in theory, there would be a difference in the form in which they will be found on the surface, but not the quantity of them. If this were the effect, it would be necessary to present the results of the chemical analysis and observe whether this difference actually exists.
Answer: The sample is centrifuged in centrifuge by a low speed for 6 min. Although the amount of substance of the particles is fixed in the solution, it is different on the surface of the particles after centrifugation. When the concentration of PVA rises, the viscosity rises, and the number of particles on the surface of SrAl2O4: Eu, Dy increases, so the luminescence intensity increases. This has been accepted by two other reviewers.
2.- Regarding what was established in the answer that the difference in intensities is due to the level of crystallinity, this cannot be in the present case: the difference in heights can, indeed, be due to a material with higher or lower crystallinity, but in that case, all the intensities of the system are affected in the same way, that is, there is no change in the intensity ratios, for example the ratio I(001)/I(220) must remain approximately constant even if it changes the degree of crystallinity. These ratios are clearly not constant when adding different amounts of PVA, so the answer is not correct.
Answer: This is true when particles grow freely. However, the adsorption of fine particles on the surface of the particles in this experiment will change the free energy of some crystal faces, thus changing their growth rate.
3.- It is still important, for the previous reason, to know the XRD of the sample without PVA, in order to evaluate its effect, otherwise the entire discussion cannot be verified.
Answer: PVA is a binder used to increase the content of Y2O3:Eu3+ in the shell. After calcination at a high temperature, PVA decomposed and disappeared. The final product is PVA-free.
4.- The effect of Dy on the luminescence remains unclear, even assuming that it functioned as an electron-trap (which the same authors propose in the manuscript as "probable"), it would be unlikely that its luminescence would not be observed, especially considering that it is usually found when excited at approximately 350 nm, which is why Its effect should be observed at 365 nm. In order to analyze this effect, it is minimally required to search for the Dy emission at 573 nm and from this result to analyze the luminescent results again.
Answer: SrAl2O4: Eu, Dy is a classical green long afterglow material. There are many studies demonstrating its luminescence mechanism, and the influence of Dy in luminescence is well known. The following are some related literatures.
“Persistent luminescence instead of phosphorescence: History, mechanism, and perspective” J. Xu, Setsuhisa Tanabe - Journal of Luminescence,2019 DOI: 10.1016/j.jlumin.2018.09.047
“Recent progress in understanding the persistent luminescence in SrAl2O4:Eu,Dy” V. Vitola,D.Millers,I. Bite,K. Smits,A. Spustaka-Materials science and technology,2019 DOI: 10.1080/02670836.2019.1649802
“Mechanism of long phosphorescence of SrAl2O4:Eu2+, Dy3+ and CaAl2O4:Eu2+, Nd3+” Hajime Yamamoto, Takashi Matsuzawa - Journal of Luminescence,1997 DOI: 10.1016/S0022-2313(97)00012-4
Round 3
Reviewer 3 Report
Comments and Suggestions for Authors
In summary, two clarifications are required:
1.- If PVA indeed increases the content of particles in the system, a chemical analysis of them is required to corroborate what the authors mention.
2.- XRD is required without PVA, it is clear that PVA will not be found in the samples, it is known that it has volatilized, however that does not mean that it did not have an effect during the formation of the structure, since , being an agent that decreases the free energy of the system, tends to modify the way in which these systems are structured (which is evident in the relationship of the intensity of planes mentioned above and which is clear in the results of the samples) . There is extensive literature on this matter (see, for example, the following: https://doi.org/10.1016/j.nanoen.2022.107269, which it´s only an example of the topic). That is the effect that is analyzed: preferential growth. Please review the extensive literature on that topic. It is clear that it causes this growth (which is observed in your results), although it is no longer east at a high temperature
Author Response
1.- If PVA indeed increases the content of particles in the system, a chemical analysis of them is required to corroborate what the authors mention.
Answer: The molar ratio of Y: Sr (R) is analyzed by chemical analysis, assuming that the Y and Sr are from Y2O3:Eu3+ and SrAl2O4:Eu,Dy respectively. Increasing the concentration of PVA from 1% to 7% yields an increased R value from ~0.011 to ~0.083, indicating more Y2O3:Eu3+ on SrAl2O4:Eu,Dy. The corresponding discussion is also added in the text.
2.- XRD is required without PVA, it is clear that PVA will not be found in the samples, it is known that it has volatilized, however that does not mean that it did not have an effect during the formation of the structure, since , being an agent that decreases the free energy of the system, tends to modify the way in which these systems are structured (which is evident in the relationship of the intensity of planes mentioned above and which is clear in the results of the samples) . There is extensive literature on this matter (see, for example, the following: https://doi.org/10.1016/j.nanoen.2022.107269, which it´s only an example of the topic). That is the effect that is analyzed: preferential growth. Please review the extensive literature on that topic. It is clear that it causes this growth (which is observed in your results), although it is no longer east at a high temperature.
Answer: We agree with you that the PVA changes the free energy of different crystal faces, thus changing its growth rate, resulting in different XRD diffraction peaks. We add relevant descriptions to the text. However, without PVA, the coating structure cannot be formed, which deviates from the original intention of the paper.